# Informationally Complete Characters for Quark and Lepton Mixings

**Michel Planat** [1,*], **Raymond Aschheim** [2], **Marcelo M. Amaral** [2] **and Klee Irwin** [2]

1   Institut FEMTO-ST CNRS UMR 6174, Université de Bourgogne/Franche-Comté,
    15 B Avenue des Montboucons, F-25044 Besançon, France
2   Quantum Gravity Research, Los Angeles, CA 90290, USA; raymond@QuantumGravityResearch.org (R.A.);
    Marcelo@quantumgravityresearch.org (M.M.A.); Klee@quantumgravityresearch.org (K.I.)
*   Correspondence: michel.planat@femto-st.fr

**Abstract:** A popular account of the mixing patterns for the three generations of quarks and leptons is through the characters $\kappa$ of a finite group $G$. Here, we introduce a $d$-dimensional Hilbert space with $d = cc(G)$, the number of conjugacy classes of $G$. Groups under consideration should follow two rules, (a) the character table contains both two- and three-dimensional representations with at least one of them faithful and (b) there are minimal informationally complete measurements under the action of a $d$-dimensional Pauli group over the characters of these representations. Groups with small $d$ that satisfy these rules coincide in a large part with viable ones derived so far for reproducing simultaneously the CKM (quark) and PNMS (lepton) mixing matrices.

**Keywords:** informationally complete characters; quark and lepton mixings; CP violation; Pauli groups

## 1. Introduction

In the standard model of elementary particles and according to current experiments, there exist three generations of matters but we do not know why. Matter particles are fermions of spin 1/2 and comprise the quarks (responsible for the strong interactions) and leptons (responsible for the electroweak interactions as shown in Table 1 and Figure 1).

**Table 1.** (1) The three generations of up-type quarks (up, charm and top) and of down-type quarks (down, strange and bottom) and, (2) the three generations of leptons (electron, muon and tau) and their partner neutrinos. The symbols $Q$, $T_3$ and $Y_W$ are for charge, isospin and weak hypercharge, respectively. They satisfy the equation $Q = T_3 + \frac{1}{2}Y_W$.

| Matter | Type 1 | Type 2 | Type 3 | $Q$ | $T_3$ | $Y_W$ |
|---|---|---|---|---|---|---|
| (1) quarks | u | c | t | 2/3 | 1/2 | 1/3 |
| | d | s | b | −1/3 | −1/2 | 1/3 |
| (2) leptons | e | $\mu$ | $\tau$ | −1 | −1/2 | −1 |
| | $\nu_e$ | $\nu_\mu$ | $\nu_\tau$ | 0 | 1/2 | −1 |

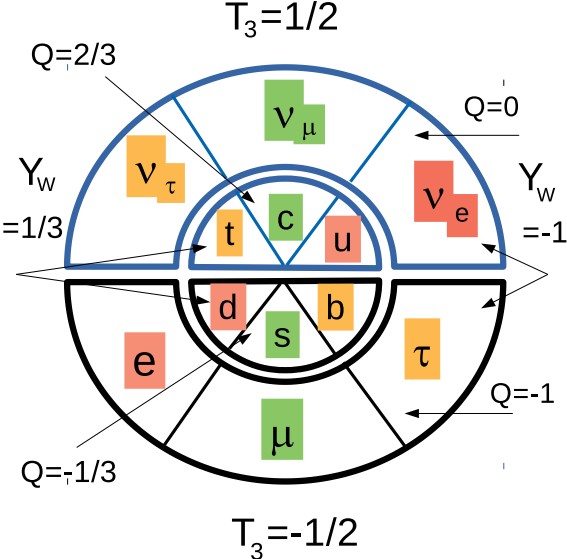

**Figure 1.** An angular picture of the three generations of quarks and leptons. The blue and black pancakes have isospin $1/2$ and $-1/2$, respectively. The inner and outer rings have weak hypercharges $\frac{1}{3}$ and $-1$, respectively.

In order to explain the CP-violation (the non-invariance of interactions under the combined action of charged-conjugation (C) and parity (P) transformations) in quarks, Kobayashi and Maskawa introduced the so-called Cabibbo-Kobayashi-Maskawa unitary matrix (or CKM matrix) that describes the probability of transition from one quark $i$ to another $j$. These transitions are proportional to $|V_{ij}|^2$, where the $V_{ij}$'s are entries in the CKM matrix [1,2]

$$U_{CKM} = \begin{pmatrix} V_{ud} & V_{us} & V_{ub} \\ V_{cd} & V_{cs} & V_{cb} \\ V_{td} & V_{ts} & V_{tb} \end{pmatrix} \text{ with } |U_{CKM}| \approx \begin{pmatrix} 0.974 & 0.225 & 0.004 \\ 0.225 & 0.973 & 0.041 \\ 0.009 & 0.040 & 0.999 \end{pmatrix}.$$

There is a standard parametrization of the CKM matrix with three Euler angles $\theta_{12}$ (the Cabbibo angle), $\theta_{13}$, $\theta_{23}$, and the CP-violating phase $\delta_{CP}$. Taking $s_{ij} = \sin(\theta_{ij})$ and $c_{ij} = \cos(\theta_{ij})$, the CKM matrix reads

$$\begin{pmatrix} 1 & 0 & 0 \\ 0 & c_{23} & s_{23} \\ 0 & -s_{23} & c_{23} \end{pmatrix} \begin{pmatrix} c_{13} & 0 & s_{13}e^{-i\delta_{CP}} \\ 0 & 1 & 0 \\ -s_{13}e^{i\delta_{CP}} & 0 & c_{13} \end{pmatrix} \begin{pmatrix} c_{12} & s_{12} & 0 \\ -s_{12} & c_{12} & 0 \\ 0 & 0 & 1 \end{pmatrix}.$$

Similarly, the charged leptons $e$, $\mu$ and $\tau$ are related to three generations of flavors of neutrinos $\nu_e$, $\nu_\mu$ and $\nu_\tau$ in the charged-current weak interaction. Neutrino mass $m_i$ can be deduced with probability $|U_{\alpha i}|^2$, where the $U_{\alpha i}$'s are the amplitudes of mass eigenstates $i$ in flavor $\alpha$. The so-called Pontecorvo–Maki–Nakagawa–Sakata unitary matrix (or PMNS matrix) is as follows [3]

$$U_{PMNS} = \begin{pmatrix} U_{e1} & U_{e2} & U_{e3} \\ U_{\mu 1} & U_{\mu 2} & U_{\mu 3} \\ U_{\tau 1} & U_{\tau 2} & U_{\tau 3} \end{pmatrix}$$

$$\text{with } |U_{PMNS}| \approx \begin{pmatrix} 0.799 \to 0.844 & 0.516 \to 0.582 & 0.141 \to 0.156 \\ 0.242 \to 0.494 & 0.467 \to 0.678 & 0.639 \to 0.774 \\ 0.284 \to 0.521 & 0.490 \to 0.695 & 0.615 \to 0.754 \end{pmatrix},$$

where the entries in the matrix mean the range of values allowed by present day experiments.

As for the CKM matrix, the three mixing angles are denoted $\theta_{12}$, $\theta_{13}$, $\theta_{23}$, and the CP-violating phase is called $\delta_{CP}$.

The current experimental values of angles for reproducing entries in the CKM and PMNS matrices are in Table 2.

**Table 2.** Experimental values of the angles in degrees for mixing patterns of quarks (in the CKM matrix) and leptons (in the PMNS matrix).

| Angles (in Degrees) | $\theta_{12}$ | $\theta_{13}$ | $\theta_{23}$ | $\delta_{CP}$ |
|---|---|---|---|---|
| quark mixings | 13.04 | 0.201 | 2.38 | 71 |
| lepton mixings | 33.62 | 8.54 | 47.2 | −90 |

Over the last twenty years, a paradigm has emerged wherein there may exist an underlying discrete symmetry jointly explaining the mixing patterns of quarks and leptons [4,5]. This assumption follows from the fact that the CKM matrix is found to be closed to the identity matrix and the entries in the PMNS matrix are found to be of order 1 except for the almost vanishing $U_{e3}$. A puzzling difference between quark and lepton mixing lies in the fact that there is much more neutrino mixing than mixing between the quark flavors. Up and down quark matrices are only slightly misaligned, while there exists a strong misalignment of charged leptons with respect to neutrino mass matrices. A valid model should account for these features.

The standard model essentially consists of two continuous symmetries, the electroweak symmetry $SU(2) \times U(1)$ (that unifies the electromagnetic and weak interactions) and the quantum chromodynamics symmetry $SU(3)$ (that corresponds to strong interactions). There are several puzzles not explained within the standard model, including the flavor mixing patterns, the fermion masses, and the CP violations in the quark and lepton sectors. There are astonishing numerical coincidences such as the Koide formula for fermion masses [6,7], the quark-lepton complementarity relations $\theta_{12}^{quark} + \theta_{12}^{lepton} \approx \pi/4$, $\theta_{23}^{quark} \pm \theta_{23}^{lepton} \approx \pi/4$ [8] and efficient first order models such as the tribimaximal model [9–12] and the "Golden ratio" model [13,14]. For instance, tribimaximal mixing gives values of angles as $\theta_{12}^{lepton} = \sin^{-1}(\frac{1}{\sqrt{3}}) \approx 35.3°$, $\theta_{23}^{lepton} = 45°$, $\theta_{13}^{lepton} = 0$ and $\delta_{CP} = 0$, compatible with earlier data. Such a model could be made more realistic by taking two CP-phases instead of one [12]. In Reference [14], the conjecture is that reality is information-theoretic as its core and the Golden Ratio is the fundamental dimensionless constant of Nature.

Currently, many discrete models of quark-lepton mixing patterns are based on the representations of finite groups that are both subgroups of $U(2)$ and $U(3)$ [15–22]. In the same spirit, we add to this body of knowledge by selecting valid subgroups of unitary groups with a criterion borrowed from the theory of generalized quantum measurements.

One needs a quantum state (called a fiducial state) and one also requires that such a state be informationally complete under the action of a $d$-dimensional Pauli group $\mathcal{P}_d$. If such a state is not an eigenstate of a $d$-dimensional Pauli group, it allows one to perform universal quantum computation [23–25]. In the above papers, valid states belong to the eigenstates of mutually commuting permutation matrices in a permutation group derived from the coset classes of a free group with relations. From here, the fiducial state will have to be selected from the characters $\kappa$ of a finite group $G$ with the number of conjugacy classes $d = cc(G)$ defining the Hilbert space dimension. Groups under consideration should obey two rules (a) the character table of $G$ contains both 2- and 3-dimensional representations with at least one of them faithful and (b) there are minimal informationally complete measurements under the action of a $d$-dimensional Pauli group over the

characters of these representations. The first criterion is inspired by the current understanding of quark and lepton mixings (and the standard model) and the second one by the theory of magic states in quantum computing [23]. Since matter particles are spin $1/2$ fermions, it is entirely consistent to see them under the prism of quantum measurements.

In the rest of this introduction we recall what we mean by a minimal informationally complete quantum measurement (or MIC). In Section 2, we apply criteria (a) and (b) to groups with small $cc \leq 36$, where we can perform the calculations. Then we extrapolate to some other groups with $cc > 36$. Most groups found from this procedure fit the current literature as being viable for reproducing lepton and quark mixing patterns. In Section 3, we examine the distinction between generalized CP symmetry and CP violation and apply it to our list of viable groups.

*Minimal Informationally Complete Quantum Measurements*

Let $\mathcal{H}_d$ be a $d$-dimensional complex Hilbert space and $\{E_1, \ldots, E_m\}$ be a collection of positive semi-definite operators (POVM) that sum to the identity. Taking the unknown quantum state as a rank 1 projector $\rho = |\psi\rangle \langle\psi|$ (with $\rho^2 = \rho$ and $\text{tr}(\rho) = 1$), the $i$-th outcome is obtained with a probability given by the Born rule $p(i) = \text{tr}(\rho E_i)$. A minimal and informationally complete POVM (or MIC) requires $d^2$ one-dimensional projectors $\Pi_i = |\psi_i\rangle \langle\psi_i|$, with $\Pi_i = dE_i$, such that the rank of the Gram matrix with elements $\text{tr}(\Pi_i \Pi_j)$, is precisely $d^2$.

With a MIC, the complete recovery of a state $\rho$ is possible at a minimal cost from the probabilities $p(i)$. In the best case, the MIC is symmetric and called a SIC with a further relation $|\langle\psi_i|\psi_j\rangle|^2 = \text{tr}(\Pi_i \Pi_j) = \frac{d\delta_{ij}+1}{d+1}$ so that the density matrix $\rho$ can be made explicit [26,27].

In our earlier references [23,24], a large collection of MICs are derived. They correspond to Hermitian angles $|\langle\psi_i|\psi_j\rangle|_{i\neq j} \in A = \{a_1, \ldots, a_l\}$ belonging to a discrete set of values of small cardinality $l$. They arise from the action of a Pauli group $\mathcal{P}_d$ [28] on an appropriate magic state pertaining to the coset structure of subgroups of index $d$ of a free group with relations.

Here, an entirely new class of MICs in the Hilbert space $\mathcal{H}_d$, relevant for the lepton and quark mixing patterns, is obtained by taking fiducial/magic states as characters of a finite group $G$ possessing $d$ conjugacy classes and using the action of a Pauli group $\mathcal{P}_d$ on them.

## 2. Informationally Complete Characters for Quark/Lepton Mixing Matrices

The standard classification of small groups is from their cardinality. Finite groups relevant to quark and lepton mixings are listed accordingly [9,15,18]. We depart from this habit by classifying the small groups $G$ of interest versus the number $d = cc(G)$ of their conjugacy classes. This motivation is due to the application of criterion (b), where we need to check whether the action of a Pauli group in the $d$-dimensional Hilbert space $\mathcal{H}_d$ results in a minimal informationally complete POVM (or MIC).

A list of finite groups $G$ according to the number of their conjugacy classes (complete only up to $d \leq 12$) is in Ref. [29]. It can also be easily recovered with a simple code in MAGMA [30]. For our application to quark and lepton mixings, we need much higher $d$. In practice, we use existing tables of subgroups of $U(3)$ (of cardinality up to 2000 in [9,15,18] and up to 1025 in [21]) to select our group candidates.

Table 3 gives the list of $16 + 2$ small groups with $cc \leq 36$ found to satisfy the two following rules: (a) the character table of $G$ contains both 2- and 3-dimensional representations with at least one of them faithful and (b) the quantum measurement is informationally complete under a $d$-dimensional Pauli group.

According to the quoted references in column 5 of Table 3, the 16 groups lead to good models for the absolute values of entries in the CKM and PMNS matrices except for the ones that have the factor $SL(2,5)$ in their signature. The two extra groups $(294, 7) = \Delta(6 \times 7^2)$ and $(384, 568) = \Delta(6 \times 8^2)$ arise when one takes into account the generalized CP symmetry, as in Section 3.

**Table 3.** List of the $16 + 2$ groups with the number of conjugacy classes $cc \leq 36$ that satisfy rules (a) and (b). As mentioned in Section 3, groups $(294, 7)$ and $(384, 568)$ need two CP phases to become viable models. The smallest permutation representation on $k \times l$ letters stabilizes the $n$-partite graph $K_k^l$ given in the fourth column. The group $\Delta(6 \times n^2)$ is isomorphic to $\mathbb{Z}_n^2 \rtimes S_3$. A reference is given in the last column if a viable model for quark and/or lepton mixings can be obtained. The extra cases with reference † and ‡ can be found in [18,21], respectively.

| Group | Name or Signature | cc | Graph | Ref. |
|---|---|---|---|---|
| SmallGroup(24,12) | $S_4$, $\Delta(6 \times 2^2)$ | 5 | $K_4$ | [15] |
| SmallGroup(120,5) | 2I, SL(2, 5) | 9 | $K_5^3$ | [20] †,‡ |
| SmallGroup(150,5) | $\Delta(6 \times 5^2)$ | 13 | $K_5^3$ | [2,15,16] |
| SmallGroup(72,42) | $\mathbb{Z}_4 \times S_4$ | 15 | $K_3^4$ | [9] |
| SmallGroup(216,95) | $\Delta(6 \times 6^2)$ | 19 | $K_6^3$ | [15] |
| SmallGroup(294,7) | $\Delta(6 \times 7^2)$ | 20 | ? | [31] |
| SmallGroup(72,3) | $Q_8 \rtimes \mathbb{Z}_9$ | 21 | $K_3^3$ | [9] |
| SmallGroup(162,12) | $\mathbb{Z}_3^2 \rtimes (\mathbb{Z}_3^2 \rtimes \mathbb{Z}_2)$ | 22 | $K_9^3$ | [2,15,18] |
| SmallGroup(162,14) | $\mathbb{Z}_3^2 \rtimes (\mathbb{Z}_3^2 \rtimes \mathbb{Z}_2)$, $D_{9,3}^{(1)}$ | 22 | $K_9^3$ | [2,15,19] |
| SmallGroup(384,568) | $\Delta(6 \times 8^2)$ | 24 | ? | [31] |
| SmallGroup(648,532) | $\Sigma(216 \times 3)$, $\mathbb{Z}_3 \rtimes (\mathbb{Z}_3 \rtimes SL(2,3))$ | 24 | ? | [15,22] |
| SmallGroup(648,533) | Q(648) , $\mathbb{Z}_3 \rtimes (\mathbb{Z}_3 \rtimes SL(2,3))$ | 24 | ? | [15,17] |
| SmallGroup(120,37) | $\mathbb{Z}_5 \times S_4$ | 25 | $K_5^4$ | † |
| SmallGroup(360,51) | $\mathbb{Z}_3 \times SL(2,5)$ | 27 | $K_{12}^6$ | † |
| SmallGroup(162,44) | $\mathbb{Z}_3^2 \rtimes (\mathbb{Z}_3^2 \rtimes \mathbb{Z}_2)$ | 30 | $K_9^3$ | [15] |
| SmallGroup(600,179) | $\Delta(6 \times 10^2)$ | 33 | $K_{10}^3$ | [2,15,16] |
| SmallGroup(168,45) | $\mathbb{Z}_7 \times S_4$ | 35 | $K_7^4$ | † |
| SmallGroup(480,221) | $\mathbb{Z}_8.A_5$, $SL(2,5).\mathbb{Z}_4$ | 36 | $K_8^6$ | ‡ |

Details are in Table 4 for the first three groups and the group $(294, 7)$. Full results are found in Tables A1 and A2 of the Appendix A.

**Table 4.** For each of the first three small groups considered in our Table 3 and the group $(294, 7)$ added in Section 3, for each character, the table provides the dimension of the representation and the rank of the Gram matrix obtained under the action of the corresponding Pauli group. Bold characters are for faithful representations. According to our requirement, each selected group has both 2- and 3-dimensional characters (with at least one of them faithful) that are fiducial states for an informationally complete POVM (or MIC) with the rank of Gram matrix equal to $d^2$. The Pauli group performing this action is a $d$-dit or a 2-qutrit (2QT) for the group $(120, 5) = SL(2, 5) = 2I$.

| Group | d | | | | | | | | | | | | | |
|---|---|---|---|---|---|---|---|---|---|---|---|---|---|---|
| (24,12) | 5 | 1 | 1 | **2** | **3** | **3** | | | | | | | | |
| 5-dit | | 5 | 21 | $d^2$ | $d^2$ | $d^2$ | | | | | | | | |
| (120,5) | 9 | 1 | **2** | **2** | 3 | 3 | 4 | **4** | 5 | **6** | | | | |
| 9-dit | | 9 | $d^2$ | $d^2$ | $d^2$ | $d^2$ | $d^2$ | $d^2$ | 79 | $d^2$ | | | | |
| 2QT | | 9 | $d^2$ | $d^2$ | $d^2$ | $d^2$ | $d^2$ | $d^2$ | $d^2$ | $d^2$ | | | | |
| (150,5) | 13 | 1 | 1 | **2** | **3** | **3** | **3** | **3** | **3** | **3** | **3** | **3** | 6 | 6 |
| 13-dit | | 13 | 157 | $d^2$ | $d^2$ | $d^2$ | $d^2$ | $d^2$ | $d^2$ | $d^2$ | $d^2$ | $d^2$ | $d^2$ | $d^2$ |
| (294,7) | 20 | 1 | 1 | 2 | **3** | **3** | **3** | **3** | **3** | **3** | **3** | **3** | **3** | **3** |
| 20-dit | | 20 | 349 | 388 | $d^2$ | $d^2$ | $d^2$ | $d^2$ | $d^2$ | $d^2$ | $d^2$ | $d^2$ | $d^2$ | $d^2$ |
| | | **6** | **6** | **6** | **6** | **6** | | | | | | | | |
| | | 390 | 390 | 390 | 398 | 398 | | | | | | | | |

Table 5 gives an extrapolation to groups with higher $cc$ where criterion (a) is satisfied but where (b) could not be checked. Most groups in the two tables have been found to be viable models, and several of them belong to known sequences.

In Tables 3 and 5, the first column is the standard small group identifier in which the first entry is the order of the group (as in [15]). In the second column, one finds the signature in terms of a direct product (with the symbol $\times$), a semidirect product (with the symbol $\rtimes$), a dot product (with the symbol .) or a member of a sequence of groups such as the $\Delta(6 \times n^2)$ sequence found to contain many viable groups for quark and lepton mixings. The third column gives the number of irreducible characters/conjugacy classes. Other information is about the geometry of the group. To obtain this geometry, one first selects the smallest permutation representation on $k \times l$ letters of $G$. Then one looks at the two-point stabilizer subgroup $G_s$ of smallest cardinality in the selected group $G$. The incidence matrix of such a subgroup turns out to be the $l$-partite graph $K_k^l$ that one can identify from the graph spectrum. Such a method is already used in our previous papers about magic state type quantum computing [23–25], where other types of geometries have been found. Finally, column 5 refers to papers where the group under study leads to a viable model both for quark and lepton mixing patterns. The recent reference [21] is taken separately from the other references singled out with the index † in the tables. It is based on the alternative concept of a two-Higgs-doublet model.

**Table 5.** List of considered groups with number of conjugacy classes $cc > 36$ that satisfy rule (a) (presumably (b) as well) and have been considered before as valid groups for quark/lepton mixing. A reference is given in the last column if a viable model for quark or/and lepton mixings can be obtained. The question mark means that the minimal permutation representation could not be obtained.

| Group | Name or Signature | $cc$ | Graph | Ref. |
|---|---|---|---|---|
| SmallGroup(726,5) | $\Delta(6 \times 11^2)$ | 38 | $K_{11}^3$ | [15,18] |
| SmallGroup(648,259) | $(\mathbb{Z}_{18} \times \mathbb{Z}_6) \rtimes S_3, D_{18,6}^{(1)}$ | 49 | $K_{18}^3$ | [2,15,18,19] |
| SmallGroup(648,260) | $\mathbb{Z}_3^2 \rtimes \text{SmallGroup}(72,42)$ | 49 | $K_{18}^3$ | [2,15,18,19] |
| SmallGroup(648,266) | $\mathbb{Z}_3^2 \rtimes \text{SmallGroup}(72,42)$ | 49 | $K_6^3$ | [15] |
| SmallGroup(1176,243) | $\Delta(6 \times 14^2)$ | 59 | $K_{14}^3$ | [15,18] |
| SmallGroup(972,64) | $\mathbb{Z}_9^2 \rtimes \mathbb{Z}_{12}$ | 62 | $K_{36}^3$ | [15,18] |
| SmallGroup(972,245) | $\mathbb{Z}_9^2 \rtimes (\mathbb{Z}_2 \times S_3)$ | 62 | $K_{18}^3$ | [18] |
| SmallGroup(1536,408544632) | $\Delta(6 \times 16^2)$ | 68 | ? | [2,15,16] |
| SmallGroup(1944,849) | $\Delta(6 \times 18^2)$ | 85 | $K_{18}^3$ | [15,18] |

## 2.1. Groups in the Series $\Delta(6n^2)$ and More Groups

An important paper dealing with the series $\Delta(6n^2) \cong \mathbb{Z}_n^2 \rtimes S_3$ as a good model for lepton mixing is [16]. A group in this series has to be spontaneously broken into two subgroups, one abelian subgroup $\mathbb{Z}_m^T$ in the charged lepton sector and a Klein subgroup $\mathbb{Z}_2^S \times \mathbb{Z}_2^U$ in the neutrino sector (with neutrinos seen as Majorana particles). The superscripts $S$, $T$ and $U$ refer to the generators of their corresponding $\mathbb{Z}_m$ group in the diagonal charged lepton basis. In this particular model, there is trimaximal lepton mixing with (so called reactor angle) $\theta_{13}$ fixed up to a discrete choice, an oscillation phase zero or $\pi$ and the (so-called atmospheric angle) $\theta_{23} = 45° \pm \theta_{13}/\sqrt{2}$.

It is shown in [2] (Table I) that two groups in this series with $n = 10$ and $n = 16$ provide leading order leptonic mixing patterns within 3-sigma of current best fit with acceptable entries in the CKM matrix. The small group $(648, 259) = D_{18,6}^{(1)}$ also satisfies this requirement. Additionally, if one accepts that neutrinos are Dirac particles, the residual symmetry group of neutrino masses is no longer restricted to the Klein group but may be any abelian group. In such a case, four small groups which are $\Delta(6 \times 5^2)$ and the small groups $(162, 10)$, $(162, 12)$ and $(162, 14) = D_{9,3}^{(1)}$ predict acceptable entries for the quark and lepton mixing matrices [2] (Table II). It is noticeable that our small selection of groups (from requirements (a) and (b)) include all of them except for the group $(162, 10)$ whose two-dimensional representations are not MICs.

Still assuming that neutrinos are Dirac particles and with loose enough constraints on $V_{us}$, paper [18] includes $\Delta$-groups with $n = 9$ (it does not lie in our Table 3) and $n = 14$ in their selection, as well as groups $(648, 259)$, $(648, 260)$ and $(648, 266)$, the latter groups are in our Table 5. Additional

material [18] provides very useful information about the ability of a group to be a good candidate for modeling the mixing patterns. According to this reference, the groups $\Delta(6 \times n^2)$ with $n = 10, 11, 14$ and 18, and small groups $(972, 64)$ and $(972, 245)$, that are in our tables, also match Dirac neutrinos with a 3-sigma fit and quark mixing patterns for triplet assignment.

Three extra groups $(120, 5)$ (the binary icosahedral group $SL(2,5) = 2I$), $(360, 51) = \mathbb{Z}_3 \times SL(2,5)$ and $(480, 221) = SL(2,5).\mathbb{Z}_4$ in our tables, whose signature has a factor equal to the binary icosahedral group $2I$, can be assigned with a doublet and a singlet for quarks but cannot be generated by the residual symmetries in the lepton sector.

### 2.2. Exceptional Subgroups of $SU(3)$

The viability of so-called exceptional groups of $SU(3)$ for lepton mixings have been studied in [22] by assuming neutrinos to be either Dirac or Majorana particles. These subgroups are listed according to the number of their conjugacy classes in Table 6. They are $\Sigma(60) \cong A_5$ (a subgroup of $SO(3)$), $\Sigma(168) \cong PSL(2,7)$, $\Sigma(36 \times 3)$, $\Sigma(72 \times 3)$, $\Sigma(360 \times 3)$ and $\Sigma(216 \times 3)$. Only group $\Sigma(360 \times 3)$ has Klein subgroups and thus supports a model with neutrinos as Majorana particles. Group $\Sigma(216 \times 3)$ is already in our Table 3 and potentially provides a valid model for quark/lepton mixings by assuming neutrinos are Dirac particles.

According to our Table 6, all these exceptional groups have informationally complete characters in regard to most of their faithful three-dimensional representations. Another useful piece of information is about groups $\Sigma(60)$ and $\Sigma(360 \times 3)$ that are informationally complete in regard to their 5-dimensional representations. Models based on the $A_5$ family symmetry are in [31,32].

**Table 6.** Exceptional subgroups of $SU(3)$. For each group and each character, the table provides the dimension of the representation and the rank of the Gram matrix obtained under the action of the corresponding Pauli group. Bold characters are for faithful representations.

| Group | $d$ | | | | | | | | | | | | | |
|---|---|---|---|---|---|---|---|---|---|---|---|---|---|---|
| (60,5), $\Sigma(60)$ | 5 | 1 | **3** | **3** | **4** | **5** | | | | | | | | |
| 5-dit | | 5 | $d^2$ | $d^2$ | $d^2$ | $d^2$ | | | | | | | | |
| (168,42), $\Sigma(168)$ | 6 | 1 | **3** | **3** | **6** | **7** | **8** | | | | | | | |
| 6-dit | | 6 | $d^2$ | $d^2$ | 33 | 33 | 33 | | | | | | | |
| (108,15), $\Sigma(36 \times 3)$ | 14 | 1 | 1 | 1 | 1 | 3 | 3 | **3** | **3** | **3** | **3** | **3** | **3** | |
| 14-dit | | 14 | 166 | 181 | 181 | 195 | 195 | $d^2$ | $d^2$ | $d^2$ | $d^2$ | $d^2$ | $d^2$ | |
| | | 4 | 4 | | | | | | | | | | | |
| | | 154 | 154 | | | | | | | | | | | |
| (216,88), $\Sigma(72 \times 3)$ | 16 | 1 | 1 | 1 | 1 | 2 | 3 | **3** | **3** | **3** | **3** | **3** | **3** | |
| 16-dit | | 16 | 175 | 175 | 157 | 233 | $d^2$ | $d^2$ | $d^2$ | $d^2$ | $d^2$ | $d^2$ | $d^2$ | |
| 2Quartits | | 16 | 121 | 149 | 125 | 200 | $d^2$ | $d^2$ | $d^2$ | $d^2$ | $d^2$ | $d^2$ | $d^2$ | |
| | | **3** | **3** | **3** | 8 | | | | | | | | | |
| 16-dit | | $d^2$ | 222 | 222 | 144 | | | | | | | | | |
| 2Quartits | | $d^2$ | 118 | 118 | 144 | | | | | | | | | |
| (1080,260), $\Sigma(360 \times 3)$ | 17 | 1 | 3 | 3 | 3 | 3 | 5 | 5 | 6 | 6 | 8 | 8 | 9 | |
| 17-dit | | 17 | $d^2$ | $d^2$ | $d^2$ | $d^2$ | $d^2$ | $d^2$ | $d^2$ | $d^2$ | $d^2$ | $d^2$ | $d^2$ | |
| | | **9** | **9** | **10** | **15** | **15** | | | | | | | | |
| | | $d^2$ | $d^2$ | $d^2$ | $d^2$ | $d^2$ | | | | | | | | |
| (648,532),$\Sigma(216 \times 3)$ | 24 | 1 | 1 | 1 | 2 | 2 | 2 | 3 | **3** | **3** | **3** | **3** | **3** | |
| 24-dit | | 24 | 527 | 527 | 562 | $d^2$ | $d^2$ | 560 | $d^2$ | $d^2$ | $d^2$ | $d^2$ | $d^2$ | |
| | | **3** | 6 | 6 | 6 | 6 | 6 | 6 | 8 | 8 | 8 | 9 | 9 | |
| | | $d^2$ | $d^2$ | $d^2$ | $d^2$ | $d^2$ | $d^2$ | $d^2$ | 564 | $d^2$ | $d^2$ | 552 | 552 | |

## 3. Generalized CP Symmetry, CP Violation

Currently, many models focus on the introduction of a generalized CP symmetry in the lepton mixing matrix [12,31,33]. The Dirac CP phase $\delta_{CP} = \delta_{13}$ for leptons is believed to be around $-\pi/2$ [33]. A set of viable models with discrete symmetries including generalized CP symmetry has been derived in [34], where full details about the so-called semidirect approach and its variant are provided. Most finite groups used for quark/lepton mixings without taking into account the CP symmetry survive as carrying generalized CP symmetries in the model described in [34]. It is found that two extra groups $(294, 7) = \Delta(6 \times 7^2)$ and $(384, 568) = \Delta(6 \times 8^2)$, that have triplet assignments for the quarks, can be added. This confirms the relevance of $\Delta$ models in this context. Group $(294, 7)$ was added to our short Table 4, where we see that all of its 2- and 3-dimensional characters are informationally complete.

We follow Reference [35] in distinguishing generalized CP symmetry from a "physical" CP violation. A "physical" CP violation is a prerequisite for baryogenesis that is the matter-antimatter asymmetry of elementary matter particles. The generalized CP symmetry was introduced as a way of reproducing the absolute values of the entries in the lepton and quark mixing matrices and, at the same time, explaining or predicting the phase angles. A physical CP violation, on the other hand, exchanges particles and antiparticles and its finite group picture had to be clarified.

It is known that the exchange between distinct conjugacy classes of a finite group $G$ is controlled by the outer automorphisms $u$ of the group. Such (non trivial) outer automorphisms have to be class-inverting to correspond to a physical CP violation [35]. This is equivalent to a relation obeyed by the automorphism $u : G \rightarrow G$ that maps every irreducible representation $\rho_{r_i}$ to its conjugate

$$\rho_{r_i}(u(g)) = U_{r_i} \rho_{r_i}(g)^* U_r^\dagger, \ \forall g \in G \text{ and } \forall i,$$

with $U_{r_i}$ a unitary symmetric matrix.

A criterion that ensures that this relation is satisfied is given in terms of the so-called twisted Frobenius-Schur indicator over the character $\kappa_{r_i}$

$$FS_u^{(n)}(r_i) = \frac{(\dim r_i)^{(n-1)}}{|G|^n} \sum_{g_i \in G} \kappa_{r_i}(g_1 u(g_1) \cdots g_n u(g_n)) = \pm 1, \ \forall i,$$

where $n = \text{ord}(u)/2$ if $\text{ord}(u)$ is even and $n = \text{ord}(u)$ otherwise.

Following this criterion, there are three types of groups

1. the groups of type I: there is at least one representation $r_i$ for which $FS_u^{(n)}(r_i) = 0$, these groups correspond to a physical CP violation,

2. groups of type II: for (at least) one automorphism $u \in G$ the $FS_u$'s for all representations are non zero. The automorphism $u$ can be used to define a proper CP transformation in any basis. There are two sub-cases:

Case II A, all $FS_u$'s are $+1$ for one of those $u$'s,

Case II B, some $FS_u$'s are $-1$ for all candidates $u$'s.

A simple program written in the Gap software allows one to distinguish these cases [35] (Appendix B).

Applying this code to our groups in Tables 3, 5 and 6, we find that all groups are of type II A or type I. Type I groups corresponding to a physical CP violation are

$$(216, 95) = \Delta(6 \times 6^2), (162, 44), (216, 88) = \Sigma(72 \times 3),$$

where we could check that our criteria (a) and (b) apply, the exceptional group $(1080, 260) = \Sigma(360, 3)$ in Table 6 and groups $(972, 64), (972, 245), (1944, 849) = \Delta(6 \times 18^3)$ of Table 5.

## 4. Conclusions

Selecting 2- and 3-dimensional representations of informationally complete characters has been found to be efficient in the context of models of CKM and PMNS mixing matrices. Generalized quantum measurements (in the form of MICs) are customary in the field of quantum information and provide a Bayesian interpretation of quantum theory leading to an innovative approach of universal quantum computing. The aim of this paper has been to see the mixing patterns of matter particles through the prism of MICs. Our method has been shown to have satisfactorily predictive power for predicting the appropriate symmetries used so far in modeling CKM/PMNS matrices and for investigating the symmetries of *CP* phases.

It is admitted that the standard model has to be completed with discrete symmetries or/and to be replaced by more general symmetries such as $SU(5)$ or $E_8 \supset SU(5)$, as in F-theory [36], to account for existing measurements on quarks, leptons and bosons, and the hypothetical dark matter. Imposing the right constraints on quantum measurements of such particles happens to be a useful operating approach.

**Author Contributions:** Conceptualization, M.P. and K.I.; methodology, M.P. and R.A.; software, M.P.; validation, R.A. and M.M.A.; formal analysis, M.P. and M.M.A.; investigation, M.P. and M.M.A.; writing–original draft preparation, M.P.; writing–review and editing, M.P.; visualization, R.A.; supervision, M.P. and K.I.; project administration, K.I.; funding acquisition, K.I. All authors have read and agreed to the published version of the manuscript.

**Funding:** Funding was obtained from Quantum Gravity Research in Los Angeles,CA

**Conflicts of Interest:** The authors declare no conflict of interest.

## Appendix A

**Table A1.** Small groups considered in our Table 3. For each group and each character, the table provides the dimension of the representation and the rank of the Gram matrix obtained under the action of the corresponding Pauli group. Bold characters are for faithful representations. According to our demands, each selected group has both 2- and 3-dimensional characters (with at least one of them faithful) that are magic states for an informationally complete POVM (or MIC), with the rank of Gram matrix equal to $d^2$. The Pauli group performing this action is in general a $d$-dit but is a 2-qutrit (2QT) for the group $(120, 5) = SL(2, 5) = 2I$, a 3-qutrit (2QT) for the group $(360, 51) = \mathbb{Z}_3 \times SL(2, 5)$ or may be a three-qubit/qutrit (3QB-QT) for the groups $(648, 532)$ and $(648, 533)$.

| Group | d | | | | | | | | | | | | | | | |
|---|---|---|---|---|---|---|---|---|---|---|---|---|---|---|---|---|
| (24,12) 5-dit | 5 | 1 5 | 1 21 | **2** $d^2$ | **3** $d^2$ | **3** $d^2$ | | | | | | | | | | |
| (120,5) 9-dit | 9 | 1 9 | **2** $d^2$ | **2** $d^2$ | 3 $d^2$ | 3 $d^2$ | 4 $d^2$ | **4** $d^2$ | 5 79 | 6 $d^2$ | | | | | | |
| 2QT | | 9 | $d^2$ | $d^2$ | $d^2$ | $d^2$ | $d^2$ | $d^2$ | $d^2$ | $d^2$ | | | | | | |
| (150,5) 13-dit | 13 | 1 13 | 1 157 | **2** $d^2$ | **3** $d^2$ | **3** $d^2$ | **3** $d^2$ | **3** $d^2$ | **3** $d^2$ | **3** $d^2$ | **3** $d^2$ | **3** $d^2$ | **6** $d^2$ | **6** $d^2$ | | |
| (72,42) 15-dit | 15 | 1 15 | 1 203 | 1 209 | 1 209 | 1 195 | 1 195 | 2 219 | 2 $d^2$ | 2 $d^2$ | 3 $d^2$ | 3 $d^2$ | **3** $d^2$ | **3** $d^2$ | **3** $d^2$ | **3** $d^2$ |
| (216,95) 19-dit | 19 | 1 19 | 1 343 | 2 357 | 2 359 | 2 355 | 2 $d^2$ | 3 $d^2$ | 3 $d^2$ | 3 $d^2$ | 3 $d^2$ | **3** $d^2$ | **3** $d^2$ | **3** $d^2$ | **3** $d^2$ | 3 $d^2$ |
| | | **3** $d^2$ | 6 $d^2$ | **6** $d^2$ | **6** $d^2$ | | | | | | | | | | | |
| (294,7) 20-dit | 20 | 1 20 | 1 349 | 2 388 | **3** $d^2$ | **3** $d^2$ | **3** $d^2$ | **3** $d^2$ | **3** $d^2$ | **3** $d^2$ | **3** $d^2$ | **3** $d^2$ | **3** $d^2$ | **3** $d^2$ | **3** $d^2$ | **3** $d^2$ |
| | | **6** 390 | **6** 390 | **6** 390 | **6** 398 | **6** 398 | | | | | | | | | | |

**Table A1.** *Cont.*

| Group | d | | | | | | | | | | | | | | | |
|---|---|---|---|---|---|---|---|---|---|---|---|---|---|---|---|---|
| (72,3) 21-dit | 21 | 1 | 1 | 1 | 1 | 1 | 1 | 1 | 1 | 1 | 2 | 2 | 2 | **2** | **2** | **2** |
| | | 21 | 405 | 405 | 421 | 421 | 421 | 421 | 421 | 421 | $d^2$ | $d^2$ | $d^2$ | $d^2$ | $d^2$ | $d^2$ |
| | | **2** | **2** | **2** | 3 | 3 | 3 | | | | | | | | | |
| | | $d^2$ | $d^2$ | $d^2$ | $d^2$ | $d^2$ | $d^2$ | | | | | | | | | |
| (162,12) 22-dit | 22 | 1 | 1 | 1 | 1 | 1 | 1 | 2 | 2 | 2 | 3 | 3 | 3 | 3 | 3 | 3 |
| | | 22 | 446 | 463 | 463 | 463 | 463 | 473 | $d^2$ | $d^2$ | $d^2$ | $d^2$ | $d^2$ | $d^2$ | $d^2$ | $d^2$ |
| | | **3** | **3** | **3** | **3** | **3** | **3** | 6 | | | | | | | | |
| | | $d^2$ | $d^2$ | $d^2$ | $d^2$ | $d^2$ | $d^2$ | 198 | | | | | | | | |
| (162,14) 22-dit | 22 | 1 | 1 | 1 | 1 | 1 | 1 | 2 | 2 | 2 | 3 | 3 | 3 | 3 | 3 | 3 |
| | | 22 | 444 | 461 | 463 | 461 | 463 | 473 | $d^2$ | $d^2$ | $d^2$ | $d^2$ | $d^2$ | $d^2$ | $d^2$ | $d^2$ |
| | | **3** | **3** | **3** | **3** | **3** | **3** | 6 | | | | | | | | |
| | | $d^2$ | $d^2$ | $d^2$ | $d^2$ | $d^2$ | $d^2$ | 198 | | | | | | | | |
| (648,532) 24-dit 3QB-QT | 24 | 1 | 1 | 1 | 2 | 2 | 2 | 3 | **3** | **3** | 3 | 3 | 3 | 3 | 6 | 6 |
| | | 24 | 527 | 527 | 562 | $d^2$ | $d^2$ | 560 | $d^2$ | $d^2$ | $d^2$ | $d^2$ | $d^2$ | $d^2$ | $d^2$ | $d^2$ |
| | | 24 | 500 | 500 | 476 | 568 | 568 | 448 | $d^2$ | $d^2$ | $d^2$ | $d^2$ | $d^2$ | $d^2$ | $d^2$ | $d^2$ |
| 24-dit 3QB-QT | | **6** | **6** | **6** | **6** | 8 | 8 | 8 | **9** | **9** | | | | | | |
| | | $d^2$ | $d^2$ | $d^2$ | $d^2$ | 564 | $d^2$ | $d^2$ | 552 | 552 | | | | | | |
| | | $d^2$ | $d^2$ | $d^2$ | $d^2$ | 448 | 560 | 560 | 510 | 510 | | | | | | |
| (648,533) 24-dit 3QB-QT | 24 | 1 | 1 | 1 | 2 | 2 | 2 | 3 | **3** | **3** | 3 | 3 | 3 | 3 | 6 | 6 |
| | | 24 | 539 | 539 | 562 | $d^2$ | $d^2$ | 514 | $d^2$ | $d^2$ | $d^2$ | 574 | 574 | $d^2$ | $d^2$ | $d^2$ |
| | | 24 | 532 | 532 | 481 | 572 | 572 | 452 | 572 | 568 | 568 | 570 | 570 | 572 | 575 | $d^2$ |
| 24-dit 3QB-QT | | **6** | **6** | **6** | **6** | 8 | 8 | 8 | **9** | **9** | | | | | | |
| | | $d^2$ | $d^2$ | $d^2$ | $d^2$ | 563 | $d^2$ | $d^2$ | 478 | 478 | | | | | | |
| | | $d^2$ | 573 | 573 | 575 | 488 | 560 | 560 | 520 | 520 | | | | | | |

**Table A2.** The following up of Table A1.

| Group | d | | | | | | | | | | | | | | | |
|---|---|---|---|---|---|---|---|---|---|---|---|---|---|---|---|---|
| (120,37) 25-dit | 25 | 1 | 1 | 1 | 1 | 1 | 1 | 1 | 1 | 1 | 1 | 2 | 2 | 2 | 2 | 2 |
| | | 25 | 601 | 601 | 601 | 601 | 601 | 601 | 601 | 601 | 601 | 623 | $d^2$ | $d^2$ | $d^2$ | $d^2$ |
| | | **3** | 3 | **3** | **3** | **3** | **3** | **3** | **3** | **3** | **3** | | | | | |
| | | $d^2$ | $d^2$ | $d^2$ | $d^2$ | $d^2$ | $d^2$ | $d^2$ | $d^2$ | $d^2$ | $d^2$ | | | | | |
| (360,51) 3QT | 27 | 1 | 1 | 1 | 2 | 2 | **2** | **2** | **2** | **2** | 3 | 3 | 3 | 3 | 3 | 3 |
| | | 27 | 613 | 613 | $d^2$ | $d^2$ | $d^2$ | $d^2$ | $d^2$ | $d^2$ | $d^2$ | $d^2$ | $d^2$ | $d^2$ | $d^2$ | $d^2$ |
| | | 4 | 4 | 4 | **4** | 4 | **4** | 5 | 5 | 5 | 6 | **6** | **6** | | | |
| | | 727 | 725 | 727 | 727 | 727 | 727 | 727 | 727 | 727 | 727 | 727 | 727 | | | |
| (162,44) 30-dit | 30 | 1 | 1 | 1 | 1 | 1 | 1 | 2 | 2 | 2 | 2 | 2 | 2 | 2 | 2 | 2 |
| | | 31 | 826 | 861 | 871 | 861 | 871 | 883 | 877 | 879 | 883 | 898 | $d^2$ | $d^2$ | $d^2$ | 898 |
| | | 2 | 2 | 2 | **3** | **3** | **3** | **3** | **3** | **3** | **3** | **3** | **3** | **3** | **3** | **3** |
| | | 898 | 898 | $d^2$ | $d^2$ | $d^2$ | $d^2$ | $d^2$ | $d^2$ | $d^2$ | $d^2$ | $d^2$ | $d^2$ | $d^2$ | $d^2$ | $d^2$ |
| (600,179) 33-dit | 33 | 1 | 1 | 2 | 3 | 3 | 3 | 3 | **3** | **3** | **3** | 3 | **3** | 3 | 3 | **3** |
| | | 33 | 1041 | $d^2$ | $d^2$ | $d^2$ | $d^2$ | $d^2$ | $d^2$ | $d^2$ | $d^2$ | $d^2$ | $d^2$ | $d^2$ | $d^2$ | $d^2$ |
| | | **3** | **3** | 3 | 3 | **3** | 3 | **6** | 6 | **6** | **6** | **6** | **6** | **6** | **6** | **6** |
| | | $d^2$ | $d^2$ | $d^2$ | $d^2$ | $d^2$ | $d^2$ | $d^2$ | $d^2$ | $d^2$ | $d^2$ | $d^2$ | $d^2$ | $d^2$ | $d^2$ | $d^2$ |
| | | **6** | 6 | **6** | | | | | | | | | | | | |
| | | $d^2$ | $d^2$ | $d^2$ | | | | | | | | | | | | |
| (168,45) 35-dit | 35 | 1 | 1 | 1 | 1 | 1 | 1 | 1 | 1 | 1 | 1 | 1 | 1 | 1 | 1 | 2 |
| | | 35 | 1175 | 1191 | 1191 | 1191 | 1191 | 1191 | 1191 | 1191 | 1191 | 1191 | 1191 | 1191 | 1191 | $d^2$ |
| | | 2 | 2 | 2 | 2 | 2 | 2 | 3 | 3 | **3** | **3** | **3** | **3** | **3** | 3 | 3 |
| | | $d^2$ | $d^2$ | $d^2$ | $d^2$ | $d^2$ | $d^2$ | $d^2$ | $d^2$ | $d^2$ | $d^2$ | $d^2$ | $d^2$ | $d^2$ | $d^2$ | $d^2$ |
| | | **3** | **3** | **3** | **3** | **3** | | | | | | | | | | |
| | | $d^2$ | $d^2$ | $d^2$ | $d^2$ | $d^2$ | | | | | | | | | | |
| (480,221) 36-dit | 36 | 1 | 1 | 1 | 1 | **2** | **2** | **2** | **2** | **2** | **2** | **2** | **2** | 3 | 3 | 3 |
| | | 36 | 36 | 1085 | 1185 | 1184 | $d^2$ | $d^2$ | $d^2$ | $d^2$ | $d^2$ | $d^2$ | $d^2$ | 1278 | 1278 | 1278 |
| | | **3** | 3 | 3 | 3 | 3 | 4 | 4 | 4 | 4 | **4** | **4** | **4** | **4** | 5 | 5 |
| | | 1278 | $d^2$ | $d^2$ | $d^2$ | $d^2$ | 1275 | 1278 | $d^2$ | $d^2$ | $d^2$ | $d^2$ | $d^2$ | $d^2$ | 1277 | 1273 |
| | | 5 | 5 | **6** | **6** | **6** | **6** | | | | | | | | | |
| | | 1294 | 1294 | 1295 | 1295 | 1295 | 1295 | | | | | | | | | |

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
