# Peer review of "Informationally Complete Characters for Quark and Lepton Mixings"

_symmetry, doi:10.3390/sym12061000_

Round 1

Reviewer 1 Report

There is a typo on line 93

Author Response

Thanks for the positive report. The referee ask the question

  1. How does this concept relate to the subject at hand?  The characters  of specific finite groups play the role of magic states to define the MICs as it is explained in Section 2. With conditions (a) and (b), a limited number of symmetries are candidates in good agreement with the litterature.
  2.  CP violation: Yes, only a few symmetries are in agreement with a so-called "physical violation". But we do not enter into particular models.

Reviewer 2 Report

The paper is written well. However, the authors can add some more references especially where they are presenting the values of the leptonic and quark mixing parameters, especially when you say,  " the Dirac cp phase is believed to be ...".

One question, if the neutrons are Majorana particles,  can you incorporate the additional phase factors for this model?.

Author Response

Thanks for the positive report. We added a missing reference after the quoted sentence.

Reviewer 3 Report

In their manuscript "Informationally complete characters for quark and lepton mixings" the authors approach the understanding of symmetries in the two mixing matrices from a fresh point of view, namely from the theory of generalized quantum measurements. As phenomenologist I understand the first part dealing with the mixing matrices but I am quite convinced that the main part of the manuscript dealing with finite groups induced by this new approach and based on previous works [24, 25] can be trusted. Still, I really would like to learn how this new approach solves such interesting puzzles like the Koide formula or the angular predictions of the tribimaximal mixing. For the latter one, I understand that this is closely related to condition (a) imposed by the authors but the connection to the former one is not explained.

In general, the manuscript deserves publication, and I am supporting this. However, I have a long list of suggestions to improve the readability of the text (and the tables). The manuscript still shows signs of being compiled from different contributions of the authors, as the phenomenological part contains more stilistic problems than others. I hope that my suggestions help to improve the manuscript.

My suggestions in chronological order (according to the line numbers):

* l. 13: interactions -> interactions)

* Table 1: in the last column replace the fullstop with a quotation mark.
In addition I suggest not to surround the table by two lines but leave the two-line style to the separation of the three horizontal blocks. The horizontal alignment should be uniformly either centered or aligned to the left or right. I advise to center the first and second block and align the rightmost block to the right in order to tabulate the signs. This comment applies also to Tables 4, 5, 7 and 8.

* Figure 1 caption: have isospin is -> have isospin
Instead of "inside and outside crowns" I suggest to use "inner and outer rings"

* l. 22: the third angle should probably be $\theta_{13}$.

* l. 24: partner wth three generations of flavors -> are related to three flavor generations

* l. 27f: PontecorvoMakiNakagawaSakata -> Pontecorco--Maki--Nakahawa--Sakata (the -- in LaTeX stands for a medium dash)

* l. 31: see line 22.

* l. 33: reproducing entries -> reproducing the entries

* Table 2: In this case the upper and lower double lines fit better than in Table 1, as they are the only ones. However, I still suggest to avoid them. In general (for all the tables in this manuscript) the caption should be separated from the table by e.g. replacing \\ by \\[7pt].

* l. 35: please skip the "or so", it appears to be slang.
it may exist -> there might exist

* l. 52f: instead of \text{quark} etc. I suggest to use {\rm quark} to make the index smaller (scriptstyle)

* l. 62: borrowed to -> borrowed from

* l. 66: When such a state -> If such a state

* l. 93: unkwown -> unknown

* l. 123: The authors use SimpleGroup from GAP. This should be cited correctly and GAP should be written in capital letters. The same is probably the case for MAGMA which, however, does not contain a procedure like SmallGroup.

* l. 126: in [21] to select our group candidates). -> in [21]) to select our group candidates.

* l. 127: Table 3 is the list -> Table 3 gives the list

* l. 128: the two rules (a) -> the following two rules: (a)

* l. 137, l. 138: are in -> are found in

* l. 139: Table 6 is -> Table 6 gives

* l. 143: tables -> Tables

* l. 144f: At the second column, one -> In the second column one

* l. 145: a signature -> the signature

* Table 3: again, please replace . -> " which is more understandable. This comment applies also to other tables.
Caption: with number -> with the number
Groups (294,7) ... as mentioned in Section 3 -> As mentioned in Section 3, the groups (294,7) ...
at the last column -> in the last column (also in caption of Table 6)

* l. 150: To get it -> To obtain this geometry

* l. 160: I am confused about whether $\dagger$ refers to [21] (as cited here) or to [22] (as cited in the Table caption).

* Table 4: Maybe the dots/fullstops can be omitted at all.
Caption: As the first sentence is not a statement, I propose to combine the two first sentences to e.g. "For each of the first three small groups considered in Table 3 and the group (294,7) added in Section 3, for each character ..."

* l. 176: particles then the -> particles, the

* l. 178: groups that are -> groups which are
and small -> and the small

* l. 181: (a) and (b) -> (a) and (b))

* l. 224: I am not sure that I understand the difference between the violation of a "generalized CP symmetry" and the "`physical' CP violation" from the phenomenological point of view. Maybe the authors want to distinguish between indirect and direct CP violations? From my point of view, both CP violations should be physical.

* l. 239: is in terms -> is given in terms

* l. 242: types of groups -> types of groups:

* l. 246: non zero -> non-zero

* l. 247: sub-cases -> sub-cases:

* l. 248: u -> $u$

* l. 300: 01301 -> 013011

* l. 327f: the comma after the last author should be attached in Refs. [21] and [22].

Author Response

Thanks for the detailed reading of the paper. We took account all the suggested corrections.

The referee asks "Still, I really would like to learn how this new approach solves such interesting puzzles like the Koide formula or the angular predictions of the tribimaximal mixing." As explained in the text, such approximations are compatible with some symmetries of the model and with condition (b), that the character is informationally complete. But there is no definitive answer.

Reviewer 4 Report

The authors  approach  nicely   the Standard model  pieces using group symmetry represenations and  characters by  performing  certain compuations. I think that this  work work will be useful for  people working on the same projects.  

Author Response

Thanks for the referee for his positive report.